# Visualizing sarcomere and cellular dynamics in skeletal muscle to improve cell therapies

Judith Hüttemeister[1,2,3], Franziska Rudolph[1], Michael H Radke[1,2], Claudia Fink[1], Dhana Friedrich[4], Stephan Preibisch[4], Martin Falcke[5], Eva Wagner[2,6], Stephan E Lehnart[2,6], Michael Gotthardt[1,2,3]*

[1]Translational Cardiology and Functional Genomics, Max Delbrück Center for Molecular Medicine, Berlin, Germany; [2]DZHK (German Centre for Cardiovascular Research), Partner Site, Berlin, Germany; [3]Charité Universitätsmedizin, Berlin, Germany; [4]Berlin Institute for Medical Systems Biology, Max Delbrück Center for Molecular Medicine, Berlin, Germany; [5]Computational Biology, Max Delbrück Center for Molecular Medicine, Berlin, Germany; [6]Heart Research Center Göttingen, Cellular Biophysics and Translational Cardiology Section, University Medical Center Göttingen, Göttingen, Germany

**Abstract** The giant striated muscle protein titin integrates into the developing sarcomere to form a stable myofilament system that is extended as myocytes fuse. The logistics underlying myofilament assembly and disassembly have started to emerge with the possibility to follow labeled sarcomere components. Here, we generated the mCherry knock-in at titin's Z-disk to study skeletal muscle development and remodeling. We find titin's integration into the sarcomere tightly regulated and its unexpected mobility facilitating a homogeneous distribution of titin after cell fusion – an integral part of syncytium formation and maturation of skeletal muscle. In adult mCherry-titin mice, treatment of muscle injury by implantation of titin-eGFP myoblasts reveals how myocytes integrate, fuse, and contribute to the continuous myofilament system across cell boundaries. Unlike in immature primary cells, titin proteins are retained at the proximal nucleus and do not diffuse across the whole syncytium with implications for future cell-based therapies of skeletal muscle disease.

*For correspondence:
gotthardt@mdc-berlin.de

Competing interest: The authors declare that no competing interests exist.

## Editor's evaluation

In this interesting study, the authors provide important insights into how titin derived from different nuclei within the syncytium is organized and integrated after cell fusion during skeletal muscle development and remodeling. This solid work elucidates the intricate process of myofilament assembly and disassembly, made possible by tracking labeled sarcomere components. The authors developed a novel mCherry titin knock-in mice with the fluorophore mCherry inserted into titin's Z-disk region to track the titin during cell fusion. The findings of the study could be important for developing therapeutic targets for diseases associated with skeletal muscle.

## Introduction

During skeletal muscle development, the first myogenic wave starts around E11 with the fusion of embryonic myoblasts at the limb buds and the dermomyotome and is accomplished by a cascade of myogenic transcription factors like myogenic factor 5 (Myf5) and myoblast determination protein (MyoD). In the second myogenic phase (E14.5–E17.5), these primary fibers fuse with fetal myoblasts

to build secondary fibers (*Chal and Pourquié, 2017*). Thereafter, some myoblasts remain less differentiated to become satellite cells, the stem cell pool in adult muscle (*Relaix et al., 2005*). They enter quiescence a few weeks after birth, and subsequently, hypertrophy is the main driver of muscle growth (*Chal and Pourquié, 2017*). In the adult, satellite cells can get activated to facilitate muscle regeneration with differentiation to myoblasts and then myocytes, which eventually undergo cell fusion to form new fibers and extend existing ones (*Almada and Wagers, 2016*).

Titin is abundantly expressed in vertebrate striated muscle (*Wang et al., 1979*), determines skeletal muscle structure and function (*Horowits et al., 1986*), and is extensively spliced to produce isoforms with differential mechanical properties (*Cazorla et al., 2000*; *Guo et al., 2012*; *Li et al., 2012*). These vary between heart and skeletal muscle and integrate into the Z-disk and M-band of the sarcomere to form a continuous elastic filament system along the myofiber (*Gregorio et al., 1998*; *Obermann et al., 1997*). The process is tightly orchestrated (*Rudolph et al., 2019*) and the resulting scaffold facilitates proper localization of sarcomeric proteins along the filament (*Rudolph et al., 2020*). Thus, traditionally, titin has been proposed to act as a molecular ruler and as a blueprint for sarcomere assembly (*Tonino et al., 2017*) and has been recognized for its role in controlling stiffness and fine-tuning contraction in mature muscle. The novelty of our study is in the ability to track titin dynamics in real time in skeletal myocytes, both in vivo and in tissue culture, during critical processes such as sarcomere remodeling and cell fusion.

With the use of fluorescent titin proteins expressed at physiological levels in knock-in mice, we have obtained insights into the titin lifecycle and sarcomere dynamics in cardiomyocytes (*da Silva Lopes et al., 2011*; *Rudolph et al., 2019*). In contrast to the heart, skeletal muscle cells form large syncytia, which contain nuclei of several fused cells. How titin moves along the large syncytium and how titins derived from different nuclei within the syncytium are organized and integrated after cell fusion has so far been prohibitively difficult to assess.

Here, we have extended the portfolio of fluorescent titin mice with the fluorophore mCherry inserted into titin's Z-disk region to follow titin not only around the sarcomere, but also during cell fusion. By using mCherry as well as eGFP knock-in mice, we can directly observe the reconstitution of the myofilament during regeneration and highlight the limitations and potential of cell-based therapies in skeletal muscle diseases. This real-time visualization of titin dynamics following cell fusion represents a significant advancement in understanding muscle development and regeneration, offering novel insights into the remodeling of the sarcomere and the mobility of titin across multinucleated syncytia, as well as the improved evaluation of cell-based therapies, as we not only learn where injected cells go, but demonstrate reconstitution of the myofilament in regenerating muscle and the limits of delivering healthy protein in a syncytium in cell-based therapy.

## Results

### The Ttn(Z)-mCherry mouse

To follow titin dynamics during cell fusion of skeletal muscle cells, we relayed on our established reporter mice, with fluorophores integrated into the M-band (*da Silva Lopes et al., 2011*) or Z-disk (*Rudolph et al., 2019*) region of titin. The knock-in approach resulted in the physiological expression of fluorescent-tagged titin and did not interfere with sarcomere assembly, titin integration, and striated muscle function. To improve the signal intensity of the red fluorophore and thus enable the analysis of skeletal muscle, we replaced dsRed at the Z-disk exon 27, C-terminal of the Z9 domain with mCherry (*Figure 1a–c*). The process involved homologous recombination in ES cells, blastocyst injection, and removal of the NEO cassette with FLP recombinase (*Figure 1a*). Homozygous and heterozygous Ttn(Z)-mCherry mice assembled functional sarcomeres with intermediate signal intensity in muscles of heterozygous mice (*Figure 1—figure supplement 1a and b*). As expected from models created earlier, there was no obvious adverse phenotype (*Rudolph et al., 2019*), no difference in heart-to-bodyweight ratio (*Figure 1—figure supplement 1c and d*), or change in titin isoform expression (*Figure 1—figure supplement 1e*), and proper co-localization of the mCherry-fluorophore with the Z-disk protein α-actinin in homozygous Ttn(Z)-mCherry and double-heterozygous Ttn(Z)-mCherry/Ttn(M)-eGFP mice (*Figure 1c and d*). Live imaging of myotubes with the SpinningDisk microscope (*Figure 1e*) confirmed an increased signal intensity of Ttn(Z)-mCherry compared with Ttn(Z)-dsRed mice (*Figure 1f*). With the improved red fluorescent label at titin's Z-disk, it is now possible to study

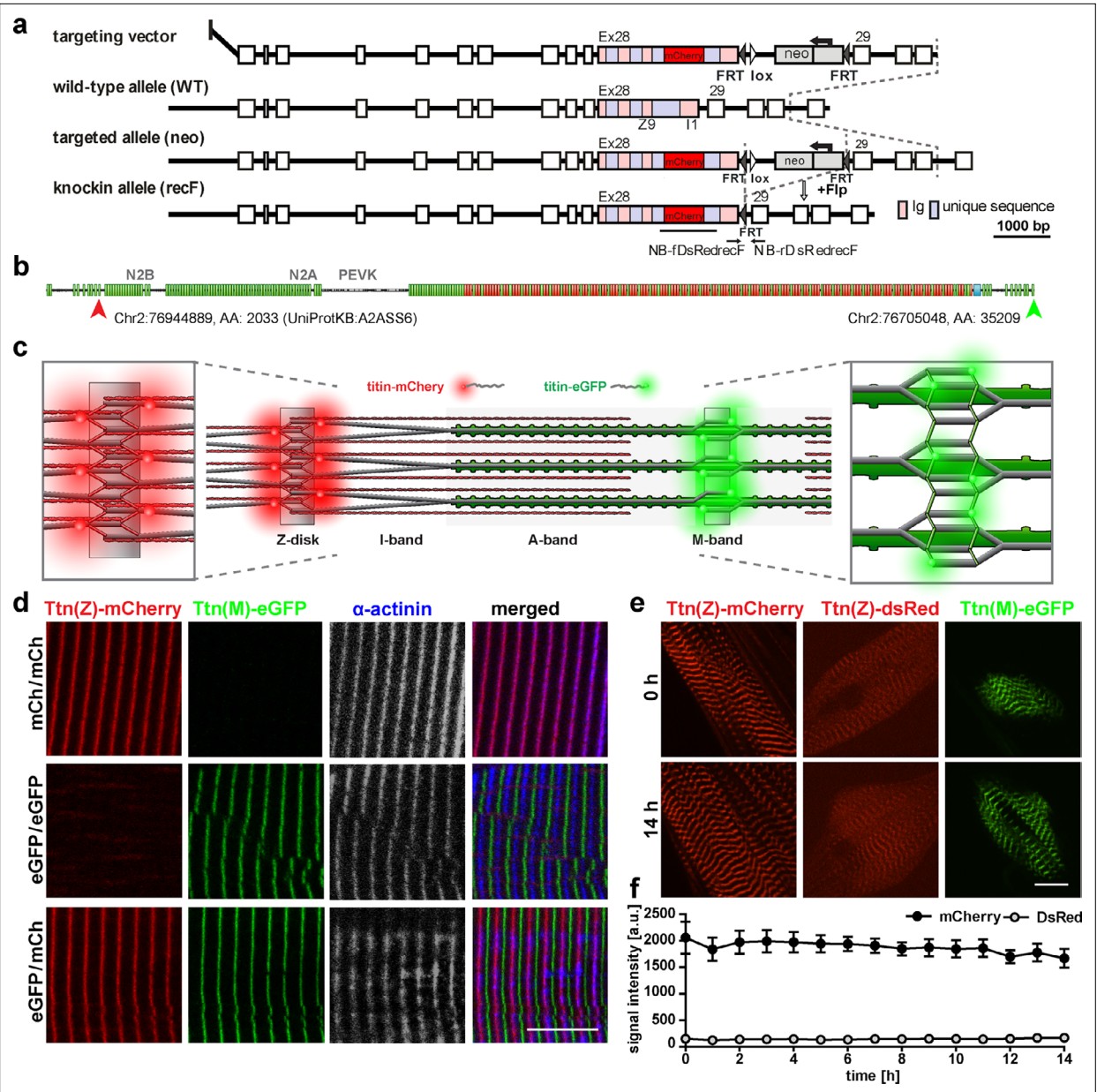

**Figure 1.** Generation and validation of a titin Z-disk knock-in with enhanced fluorescence. (**a**) Targeting strategy to insert *mCherry* into *titin*'s Exon 28 (Z-disk). (**b**) *mCherry* is integrated outside the Z9 domain (red arrow) and GFP at the c-terminus (green arrow). (**c**) Sarcomeric structure with titin-mCherry integration flanking the Z-disk and titin-GFP at titin's C-terminus at the M-band. (**d**) Alternating red and green fluorescent staining in tibialis anterior (TA) of homozygous titin(Z)-mCherry, homozygous titin(M)-eGFP, and double-heterozygous mice. Co-staining for α-actinin as a marker of the Z-disk confirms a proper localization of the mCherry fluorophore. (**e**) Simultaneous live imaging of myotubes with dsRed or mCherry fused to titin reveals higher intensity and better signal-to-noise ratio for the mCherry fluorophore. Scale bar 10 µm. (**f**) Stability of the fluorescent signal with minor changes over 14 hr and higher intensity of the mCherry signal measured in three cells (n = 3) as the average signal at three Z-disks. Difference is significant with p<0.001 for all time points, tested by two-way ANOVA.

The online version of this article includes the following source data and figure supplement(s) for figure 1:

**Source data 1.** Measured signal intensity of mCherry and dsRed from 0 to 14 hr depicted in panel f.

**Figure supplement 1.** Knock-in of mCherry at titin's Z-disk.

**Figure supplement 1—source data 1.** File with values for measured signal intensity at the Z-disk of mCherry in EDL muscle of homozygote (tg/tg), heterozygote (tg/wt), and wildtype (wt/wt) animals, corresponding to panel b; bodyweight (g), corresponding to panel c; heart weight and HW/BW ratio corresponding to panel d.

**Figure supplement 1—source data 2.** Original titin gel for panel e indicating the relevant bands of titin (N2A, T2) and MHC of wildtype (+/+),

*Figure 1 continued on next page*

*Figure 1 continued*

heterozygote (+/m), and homozygote (m/m) mCherry skeletal muscle.

**Figure supplement 1—source data 3.** Original file for titin gel displayed in panel e.

the dynamics of endogenously expressed titin simultaneously at Z-disk and M-band and even in immature myocytes during cell fusion.

## Titin kinetics in double-heterozygous myotubes

Measurements of titin kinetics in cardiomyocytes revealed that titin is not a static backbone, but dynamically exchanged in the sarcomere within hours with a faster exchange rate at its Z-disk region (*da Silva Lopes et al., 2011*; *Rudolph et al., 2019*). The different cell morphology and titin isoform composition between heart and skeletal muscle prompted the question whether titin kinetics is different in skeletal muscle cells, which we addressed using fluorescence recovery after photobleaching (FRAP) in Ttn(Z)-mCherry/Ttn(M)-eGFP double-heterozygous myotubes. In the representative images, the Ttn(Z)-mCherry signal reemerges already after 1 hr as compared to 4 hr for the Ttn(M)-eGFP signal and documented in the respective line profiles (*Figure 2a*). To confirm that the recovery of the fluorescence signal is due to titin protein exchange and not caused by a reactivation of the fluorophore, we performed the same experiment in fixed cells, where the striated signal pattern did not recover (*Figure 2—figure supplement 1a*). Only minimal background fluorescence was recovered in fixed cells after 14 hr with no difference between Ttn(Z)-mCherry and Ttn(M)-eGFP (*Figure 2—figure supplement 1b–d*). In contrast, there was a significant difference in fluorescence recovery and hence protein exchange between mCherry-labeled Z-disk titin and eGFP-labeled M-band titin in living cells (*Figure 2b*). The mobile fraction of Z-disk titin is significantly higher than the mobile fraction of M-band titin with 73 vs. 46% (*Figure 2c*), although there is variability between individual cells. The faster recovery of the mCherry-titin signal is also reflected in its significantly reduced exchange half-life of 1.5 hr compared to the 4.9 hr for the Ttn(M)-eGFP signal (*Figure 2d*). To the average fluorescence recovery (*Figure 2b*) as well as for the recovery in most individual cells, a two-phase association curve provided a better fit to the data points than the classical one-phase association curve, suggesting that the measured signal can be attributed to two protein isoform populations with different kinetics. The percentage of the fast population is significantly higher for Ttn(Z)-mCherry than for Ttn(M)-eGFP with 37 vs. 16% (*Figure 2e*). Quantification of the fluorescence signal at the opposite ends of the half-sarcomere (red signal at the M-band and green signal at the Z-disk) allowed us to quantify the kinetics of nonintegrated titin. Outside their respective integration sites, there was no significant difference anymore between the recovery of mCherry-labeled Z-disk region and the eGFP-labeled M-band region of titin (*Figure 2f*, *Figure 2-figure supplement 1e*). However, while there was no difference in mobile fraction and ratio of slow to fast population, there was still a significant difference in exchange half-life (*Figure 2—figure supplement 1f–h*). There was no significant difference between integrated and nonintegrated Z-disk titin (determined at its integration site and between, respectively), but there was an increased fluorescence recovery of nonintegrated titin-eGFP (significant from 6 to 10 hr). Of note, the nonintegrated titin signal was much lower than the signal at the integration sites. It appears that titin exchange kinetics in skeletal muscle myotubes are faster at titin's Z-disk vs. its M-band with similar rates as in embryonic cardiomyocytes (*Rudolph et al., 2019*), although the cells are structural different and contain different titin isoforms.

## Sarcomeric protein dynamics after cell fusion

A remarkable feature of skeletal muscle cells is that they form large, multinucleated syncytia arising from cell fusion. It is not completely understood so far how sarcomeric proteins of different ancestor cells are distributed and integrated along the myotube.

To address these questions, we co-cultured myoblasts of homozygous Ttn(M)-eGFP and homozygous Ttn(Z)-mCherry mice at high density and differentiated them by withdrawing growth factors 1 day later for 2–3 days to induce their fusion. After fixation, we found cells at different states of differentiation (*Figure 3*). In the first phase of fusion, cells had made initial contact as determined by visualizing cell contact formation with M-cadherin staining (*Figure 3—figure supplement 1a*), but titin-eGFP and mCherry-titin proteins had not mixed yet (*Figure 3a*), suggesting that membrane

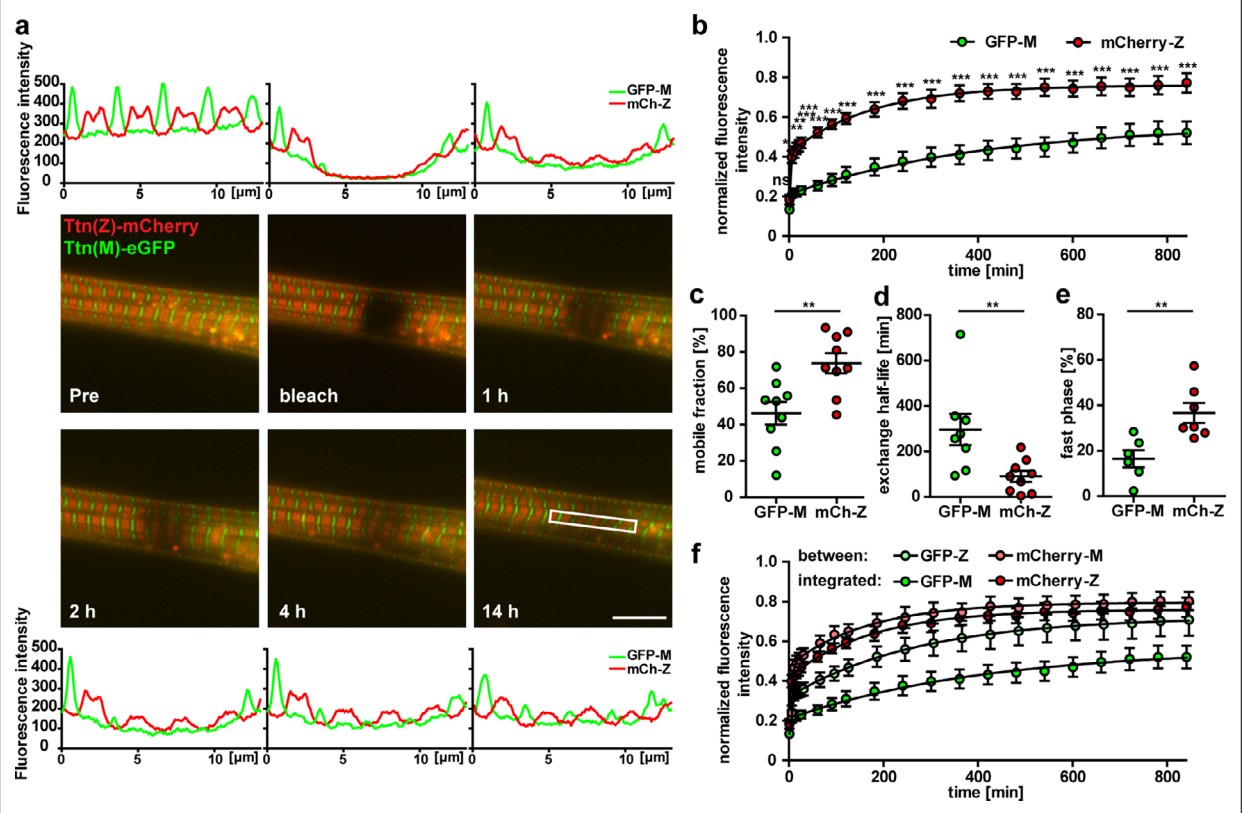

**Figure 2.** Titin mobility in myotubes at Z-disk and M-band. (**a**) Representative recovery of the sarcomeric titin signal within 14 hr. Intensity profiles of the bleached regions (white rectangle). Scale bar 10 µm. (**b**) The mCherry-labeled Z-disk titin (mCh-Z) recovers significantly faster than the GFP-labeled M-band titin (GFP-M; n = 9; in three independent experiments, two-way ANOVA) with mobile fraction increased (**c**) and exchange half-life reduced (**d**). (**e**) The recovery of fluorescent titin is biphasic with a higher contribution of the fast phase for Z-disk vs. M-band titin. In the fast phase, the signal recovers within <5 min, the slow phase lasts up to 12 hr. (**c–e**) n = 6–9 cells per group exclusion of one outlier in (**d**) and of cells without a biphasic recovery in (**e**); one-way ANOVA for (**c, d**) and (**e**). (**f**) Nonintegrated GFP-labelled titin (GFP signal outside the M-band) recovers faster than M-band integrated GFP-titin. Samples with a obvious decrease in cell quality during imaging were excluded from the analysis.

The online version of this article includes the following source data and figure supplement(s) for figure 2:

**Source data 1.** Measured intensity of mCherry-Z and GFP-M intensity at different timepoints from prebleached to 8 hr post bleaching, corresponding to panel a.

**Source data 2.** Values of normalized intensity of mCherry-Z and GFP-M signal from 0 to 840 min of recovery after bleaching, corresponding to panel b.

**Source data 3.** Values of GFP-M and mCh-Z of mobile fraction and mobile fraction (in %) corresponding to panel c, exchange half-life (min) corresponding to panel d, and fast phase (in %) corresponding to panel e.

**Source data 4.** Values of normalized fluorescence intensity of integrated (GFP-M and mCh-Z) and between (GFP-Z and mCh-M) from 0 to 840 min of recovery after bleaching, corresponding to panel f.

**Figure supplement 1.** Titin kinetics in double-heterozygous myotubes.

**Figure supplement 1—source data 1.** File with values of normalized intensities from 0 to 8 hr of GFP-M and mCherry-Z signal in fixed cells, corresponding to panel b; normalized intensity of mCherry-Z signal in live and fixed cells, corresponding to panel c; normalized intensity of GFP-M signal in live and fixed cells, corresponding to panel d; normalized intensity of mCherry-M and GFP-Z signal, corresponding to panel e; values of mobile fraction (in %) corresponding to panel f; exchange half-life (min) corresponding to panel g; two-phase recovery (in%) corresponding to panel h.

breakdown had not happened. Other cells had already fused as differentially labeled titin had started to mix (*Figure 3b*). Here, the alternating mCherry and eGFP signals in the central region of the syncytium indicate the proper integration of titin protein originating from different nuclei. The lower region contained mainly mCherry-titin, suggesting that the lower nucleus originated from a Ttn(Z)-mCherry homozygous myocyte. In some syncytia, titin had already distributed completely (likely an early fusion event), so that the nuclei could have originated from either background (*Figure 3c*). Sarcomeric proteins such as α-actinin are present (*Figure 3—figure supplement 1b and c*) and localize toward their position in the newly formed sarcomeres throughout the cell (*Figure 3—figure supplement 1c*).

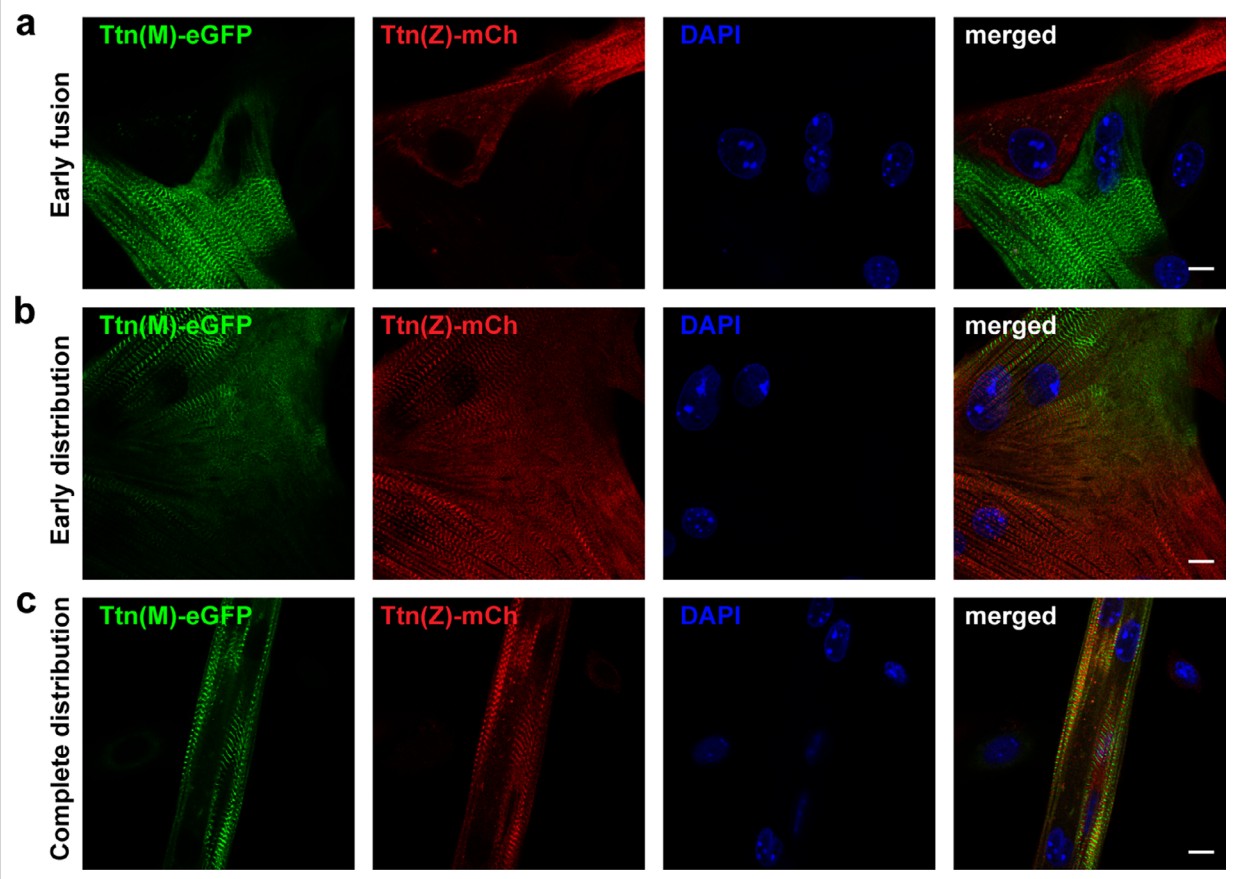

**Figure 3.** After cell fusion, titin is distributed throughout the myotube. Satellite cells were isolated from homozygous titin(Z)-mCherry and titin(M)-eGFP mice. After co-cultivation and differentiation to myotubes, cells were fixed at different stages of cell fusion, from first contact and early fusion (**a**) to early (**b**) and late (**c**) distribution of titin proteins. Scale bar 10 μm, experiment was replicated on 4 days with at least five images per staining (over 30 images of fusion events in total).

The online version of this article includes the following figure supplement(s) for figure 3:

**Figure supplement 1.** Titin distribution after cell fusion.

## Following titin along the syncytium in real time

To follow the progression of cell fusion and titin distribution, we acquired time lapses from 4 to 6 hr after initiating differentiation for 16 hr total. We successfully recorded several fusion events with myotubes of different sizes fused in different orientations (cell-to-cell or perpendicular). As determined by the sarcomere structure, we documented fusion events between two immature cells or between an immature cell and a mature cell/myotube. We followed the localization of nuclei expressing red or green titin in the syncytium and quantified the distribution of titin over time (*Figure 4*). The area where both titin signals were present above threshold levels was subdivided into areas with mainly red signal, mainly green signal, and an area with similar amounts of red and green titin. We also provide a movie to follow the fusion event in a time lapse (*Figure 4—video 1*).

In *Figure 4a*, two fusion events are indicated with white arrows directed at the points of contact. The first fusion event at 0 hr of an eGFP myocyte with a large mature multinucleated mCherry myotube leads to the gradual diffusion of eGFP-titin that ultimately contributes to <50% of the sarcomeres (*Figure 4b*). The second fusion event at 5.5 hr, two small immature cells fuse, followed by the rapid distribution of mCherry-titin and titin-eGFP. Within 1 hr, about 90% of the area is occupied by titins from both original cells (*Figure 4c*). For statistical validation of the increased speed of titin distribution in cells fusing to immature vs. mature myotubes, we quantified 9 fusion events out of 13 captured. To minimize the effects of size differences of the syncytia, we excluded very small (<1000 μm$^2$) and very large (>10,000 μm$^2$) cells. As there was still a trend for cells with a mature sarcomere structure

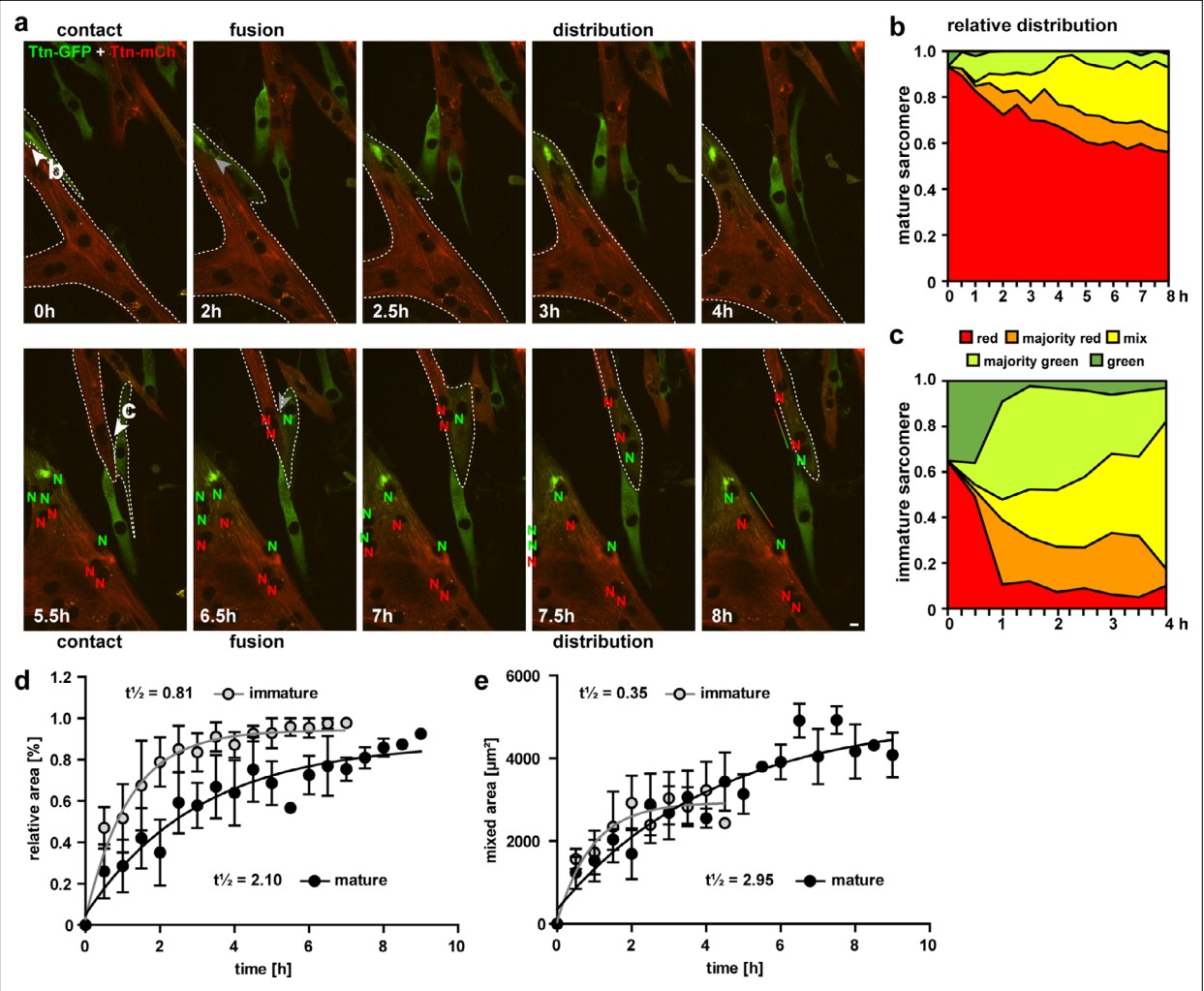

**Figure 4.** Cell fusion and redistribution of red and green titin in skeletal muscle cells. (**a**) Live imaging of cell fusion of homozygous Ttn(Z)-mCherry and Ttn(M)-eGFP myogenic cells (two frames per hour). Arrows indicate the initiation of cell fusion between a small Ttn-eGFP myocyte and a mature myofiber (**b**) compared to the fusion of two small, immature cells (**c**). Nuclei expressing red and green fluorescent titin are labeled with red and green 'N', respectively. Regions with different fluorescent titin ratios are indicated with dashed outlines. Gradient bar (8 hr) indicates the range of titin spread between two neighboring nuclei. Scale bar 10 μm. (**b, c**) Titin distribution is measured as the area containing red only, majority red, even mix, majority green, or green only based on thresholds set at 20 and 50% maximal fluorescence intensity as detailed in 'Methods'. (**b**) Fusion between a green immature and large red mature cell leads to a gradual redistribution of green titin to less than half of the resulting syncytium within 8 hr. (**c**) Fusion between a red and green immature cell leads to a rapid redistribution within the first hour that is almost complete by 4 hr. Relative (**d**) and absolute (**e**) increase of the area with mixed red and green titins in immature cells fusing with mature cells (black) vs. immature cells (gray) indicate a >2.5× faster titin distribution when both cells are immature (n = 5 for only immature cells and n = 4 fusion events with one mature cell), two-way ANOVA.

The online version of this article includes the following video and source data for figure 4:

**Source data 1.** Area occupied by fluorophore (red and green) from 0 up to 4 or 8 h of myotube maturation, corresponding to panel b and immature myotubes corresponding to panel c.

**Source data 2.** Values of mature and immature cells from 0 to 8 hr of relative area (%), corresponding to panel d, and absolute mixed area, corresponding to panel e.

**Figure 4—video 1.** Fusion movie: time lapse of cell fusion.

https://elifesciences.org/articles/95597/figures#fig4video1

to be larger, we provide relative (*Figure 4d*) and absolute values (*Figure 4e*), with titin mobility (t1/2) reduced by more than twofold in immature cells undergoing fusion.

## Titin mRNA localization after cell fusion

To dissect the contribution of titin mRNA vs. protein to titin mobility along the syncytium, we visualized titin mRNA originating from different myocytes using smFISH with probes directed against GFP (labeled with Quasar570) and mCherry mRNA (labeled with Quasar670). Homozygous Ttn(Z)-mCherry and Ttn(M)-eGFP cells were plated together and differentiated to induce cell fusion. The captured images of these experiments contain five channels (*Figure 5a*): nuclei stained with DAPI (blue), Titin-eGFP protein (green), *Ttn*-eGFP mRNA (red 570), mCherry-titin protein (red 610), and *Ttn*-mCherry mRNA (far red).

Dots representing RNA signal were most intense in the nuclei and correspond to the transcription sites of titin (two main dots for two chromosomes in *Figure 5b*). The nuclei contain only the mRNA from the cell they originated from, as confirmed by the strict separation of nuclear *Ttn*-mCherry or *Ttn*-eGFP mRNA. The myotube in the representative image of *Figure 5b* had four nuclei with *Ttn*-eGFP (first row) and five nuclei with *Ttn*-mCherry mRNA (second row), summarized in the schematic overview above the image panel. The signal dots from *Ttn*-mCherry RNA appear much more intense than from *Ttn*-eGFP RNA signal dots and could relate to the insertion of mCherry at the 5′ end of the *titin* mRNA, which leads to an earlier transcription as compared to eGFP, inserted at the 3′ end. In the myotube in *Figure 5b*, the titin proteins of different origin were not distributed completely over the whole syncytium (last row), indicating that fusion had just started. Therefore, there are still areas with mainly Ttn-eGFP protein (*Figure 5b* magnification 1) or more mCherry-titin protein (magnification 2). In these areas, we also found *titin* mRNA of both species, with mRNA from the distant nucleus underrepresented (e.g., *titin*-eGFP signal dots in the second magnification). In myotubes at a later stage after fusion with completely distributed titin protein (representative image in *Figure 5—figure supplement 1a*), *titin* mRNAs of both origins were present at the edge of the cell (magnification). These data indicate that it is not only titin protein that is distributed through the syncytium after cell fusion, but also *titin* mRNA. We compared the distribution of *titin* RNA to the titin protein distribution by measuring the relative area of mCherry and eGFP populations (*Figure 5—figure supplement 1b and c*). The size of these overlap regions varies a lot between cells depending on how far along in the fusion process the cells are. Nevertheless, there is a significant correlation between the size of the protein overlap region and the RNA overlap region (*Figure 5—figure supplement 1b*). Thereby, the RNA overlap region is significantly smaller than the protein overlap region (after correction for matched values, *Figure 5—figure supplement 1c*, left diagram).

## A theoretical approach to titin protein localization after cell fusion

We assume that red (green) titin is produced in the red (green) area and diffuses into the green (red) area while decaying according to the rate causing its half-life. The titin half-life in cultured skeletal muscle cells from day 12 chicken embryos is about 70 hr (*Isaacs et al., 1989*). In the adult mouse heart, tamoxifen induction of the conditional titin knockout leads to a maximum of ~55% truncated titin after 80 days and ~30% truncated titin after 5 days (*Peng et al., 2007*), suggesting a half-life of adult cardiac titin between 4 and 5 days (100–120 hr). Based on the embryonic chicken skeletal muscle and adult mouse heart data, we conservatively estimate the titin half-life at 3.5 days ($\tau = 3.5$, $d = 3.024 \times 10^5$ s). We estimated the titin diffusion coefficient D as 0.3 $\mu m^2$ $s^{-1}$. The spatial decay length in a diffusion profile is $(D\tau/0.693)^{1/2}$. The measured width of the titin gradient is $d = 50$ $\mu m$ (*Figure 4*, 8 hr), which is not compatible with the $\tau$ and D values. If we accept the value for D, the value of $\tau$ required to explain this width is $0.693d^2/D = 1732.5$ $\mu m^2/0.3$ $\mu m^2$ $s^{-1} = 5775$ s (<100 min), that is, unrealistically short. If we accept the $\tau$-value of 3.5 days, the diffusion coefficient to explain the gradient would be $D = 0.693d^2/\tau = 5.7 \times 10^{-3}$ $\mu m^2$ $s^{-1}$, that is, two orders of magnitude smaller than the value determined in cultured cells. Hence, another mechanism must act to restrict titin protein spread.

## Titin mobility and integration after in vivo regeneration and cell transplantation

The fusion of cultured myoblasts to multinucleated myocytes is a model for critical milestones in the development and regeneration of skeletal muscle. However, regeneration in vivo requires additional

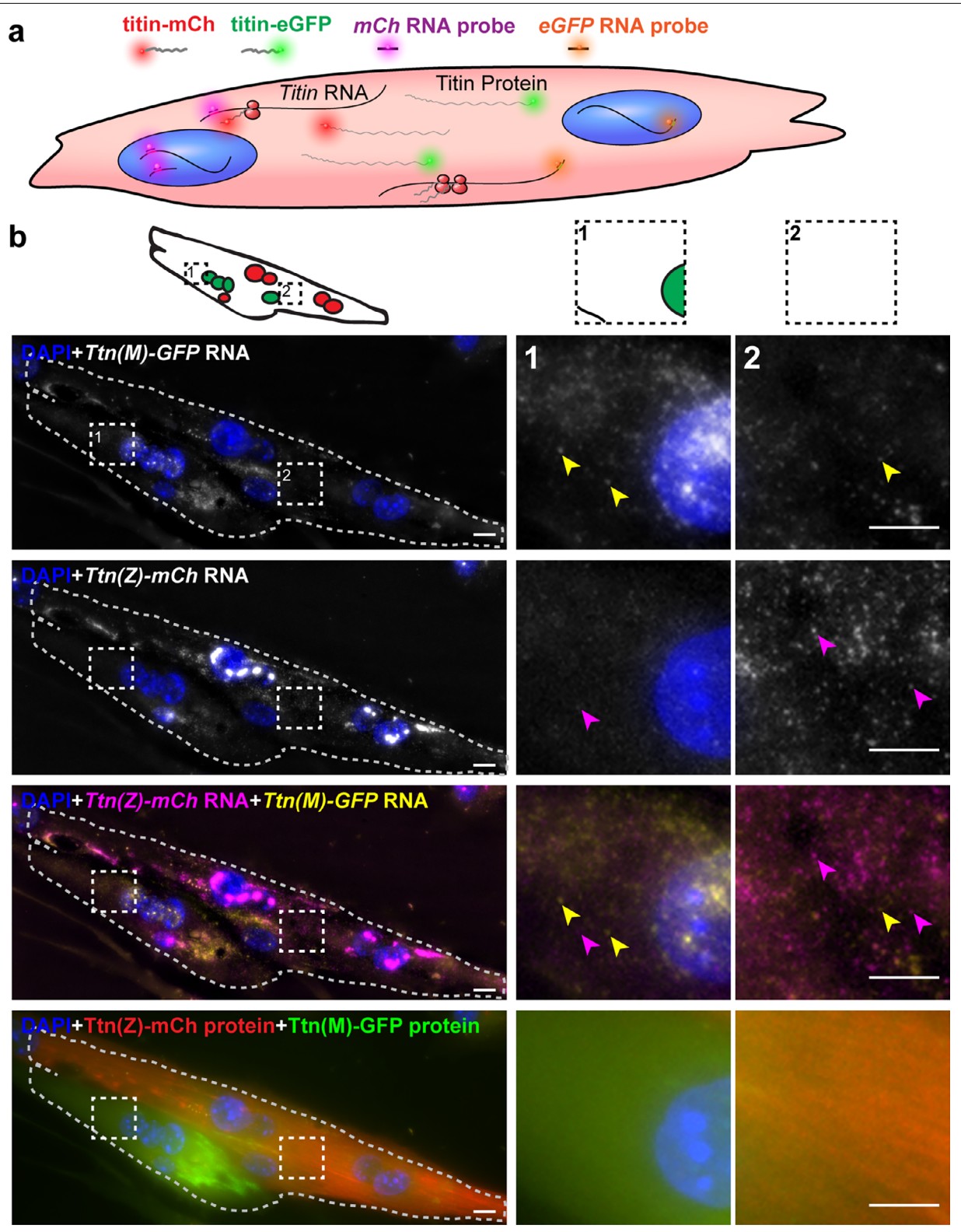

**Figure 5.** Distribution of titin mRNA in skeletal muscle cells undergoing cell fusion determined by smFISH detecting mCherry and GFP coding region. (a) Model of titin mRNA and protein synthesis and localization. (b) Representative image of a fusing myotube, where the distribution of titin mRNA has just started. While we find both titin-mCh and titin eGFP mRNA in both the red and green compartments after cell fusion (magenta and yellow arrows), we do not see crossover of green and red titin fusion protein in the same area. Scale bar 10 µm, we replicated the experiment on 2 days with 11 images of fusion events.

*Figure 5 continued on next page*

*Figure 5 continued*

The online version of this article includes the following source data and figure supplement(s) for figure 5:

**Source data 1.** File with values for mRNA and protein of cell fusion events, corresponding to panel b.

**Figure supplement 1.** Titin mRNA localization after cell fusion.

**Figure supplement 1—source data 1.** File with values for RNA and protein overlap correlation of fusion events, corresponding to panel b, and signal localization/overlap of RNA protein of mCherry and GFP in fusion events corresponding to panel c.

important steps and players, such as immune cells and the extracellular matrix. At the final stages of skeletal muscle formation in vivo, myotubes have formed muscle fibers, which are further differentiated and much larger than the myotubes that form in vitro. To evaluate if cell fusion provides additional benefits in animal experiments with cell transplantation (*Darabi et al., 2012*), we studied whether titin proteins from donor cells were distributed and integrated into the sarcomere lattice in vivo. Accordingly, we isolated donor myoblasts from Ttn(M)-eGFP mice and injected them into the tibialis anterior (TA) muscle of Ttn(Z)-mCherry mice 1 day after injury and induction of regeneration with cardiotoxin (CTX) (experimental design, *Figure 6a*). Control groups received only CTX or only cell transplantation, respectively. After 3 weeks of regeneration, we dissected the treated and untreated contralateral TA muscles and cut them in half for longitudinal and transversal cryosections.

The injection of CTX caused muscle degeneration, followed by regeneration that was largely completed after 3 weeks, when individual regenerating cells were still present – as determined by their centralized nucleus (DAPI and laminin staining; *Figure 6b and d*, *Figure 6-figure supplement 1a and c*). In the control group with CTX only (*Figure 6—figure supplement 1a*) and the mice with CTX and cell injection (*Figure 6b*), the fibers contain mainly these centralized nuclei, suggesting a successful completion of the degeneration–regeneration cycle. In the control group with only myoblast injection, only a very few myofibers contained centralized nuclei, located directly at the injection site (*Figure 6d*). The untreated contralateral muscles had no fibers with central nuclei (*Figure 6—figure supplement 1c*).

Successful transplantation of the Ttn(M)-eGFP myoblasts was detected in transversal sections with eGFP-positive fibers in the injured area (*Figure 6b*). In longitudinal sections, we confirmed the proper integration of titin protein of the donor cells into the Ttn(Z)-mCherry muscle by the periodic staining of the myofilament in longitudinal sections (*Figure 6c*). At several sites, muscle fibers were eGFP-positive (green arrows, magnifications in *Figure 6—figure supplement 1e*), as transplanted Ttn(M)-eGFP cells had differentiated together with the endogenous Ttn(Z)-mCherry satellite cells to mature muscle fibers. However, titin-eGFP signal was not evenly distributed over the complete fiber, but remained mainly proximal to the grafted nucleus.

Interestingly, in the control group with cell transplantation only (without prior injury) eGFP-positive fibers were present at the injection site (*Figure 6d*). Some of these fibers were also mCherry positive – primarily located along the injection canal. This finding is consistent with the insertion of the needle-activating endogenous satellite cells, which subsequently fused with the transplanted eGFP myoblasts (*Figure 6d*). In the control with CTX injury only (*Figure 6—figure supplement 1b*) and in the contralateral muscles (*Figure 6-figure supplement 1d*), eGFP-positive fibers were absent.

In summary, cell transplantation can be used to deliver sarcomeric proteins to regenerating muscle. Without prior injury cells remained at the injection site (*Figure 6d and e*), but in injured muscle donor cells distribute over a much larger area (*Figure 6b and c*). Fusion events can either lead to homogeneous distribution of red and green titin molecules if diffusion in vivo was not limited or eGFP and mCherry-titin stay around the respective nucleus if diffusion was limited. This is the case for titin as demonstrated in eight out of eight in vivo fusion events depicted in *Figure 6* and *Figure 6-figure supplement 1*. Here, titin travels only in a limited area around the donor nucleus even after 3 weeks, so that a sarcomeric protein would be more confined to the fusion site vs. the benefit of distributing the therapeutic protein over the whole syncytium.

## Discussion

Myofilament remodeling and adaptation are critical to balance efficient force generation and muscle mass. This includes how sarcomeres are formed, fortified, integrated into larger functional units,

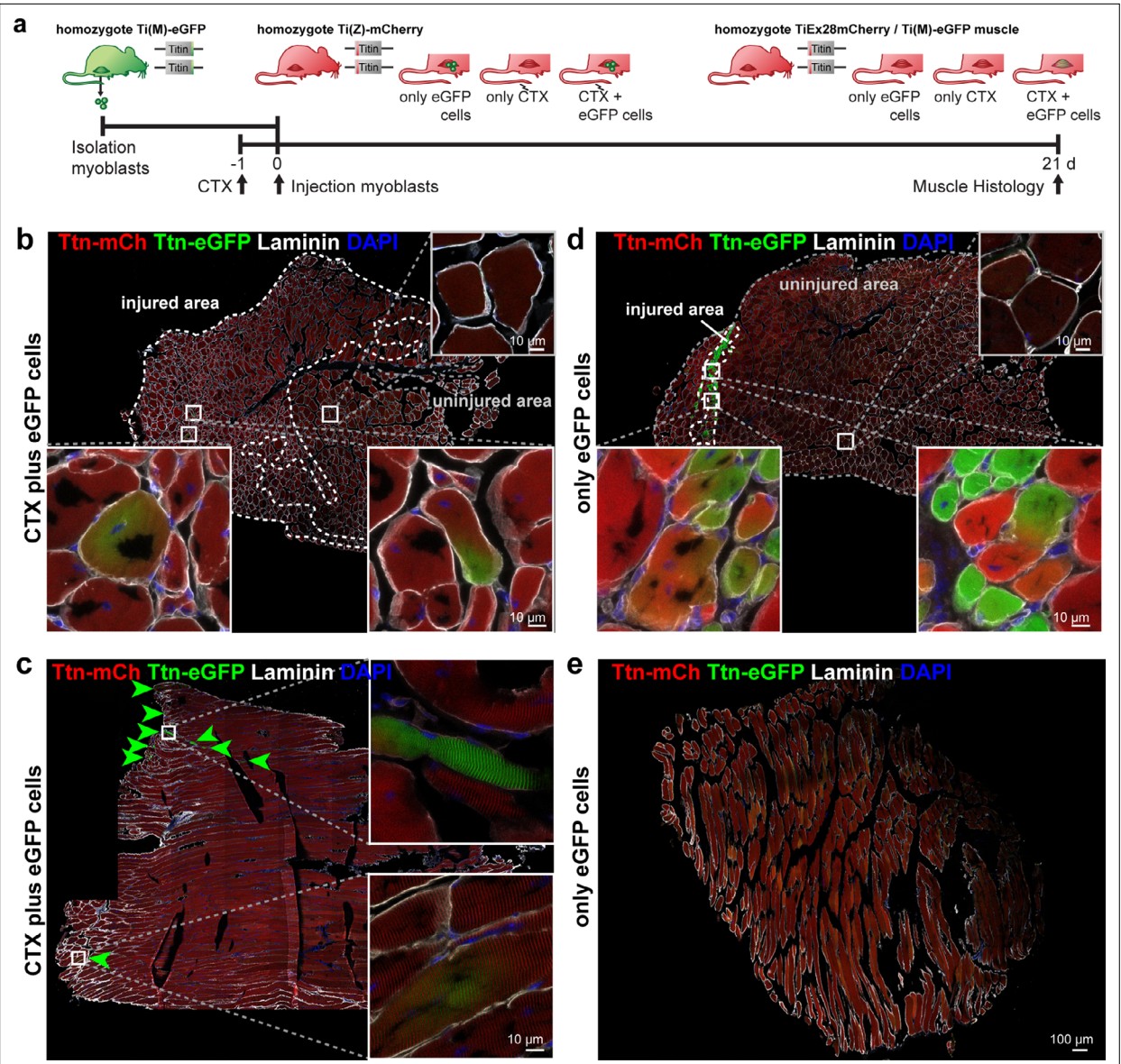

**Figure 6.** Titin distribution during regeneration. (**a**) Tibialis anterior (TA) muscle of Ttn(Z)-mCherry mice was injured by injection of cardiotoxin (CTX) followed by transplantation of Ttn(M)-eGFP myoblasts on the following day (n = 3). Controls comprise CTX injury only (n = 1) and eGFP cell transplantation without injury (n = 4). After 3 weeks of regeneration, muscles (treated and untreated contralateral TA) were dissected and sections stained against laminin to visualize cell boundaries. Centralized nuclei in transversal sections (**b, d**) are a sign of regenerating cells within injured areas. These areas contain GFP-positive fibers and extend throughout the muscle after CTX injury (**b**), but are limited to the injection site with cell injection only (**d**). The longitudinal sections (**c, e**) provide additional information about the proper integration of titin proteins from the transplanted cells into the regenerating myofibers. After injury, transplanted Ttn(M)-eGFP myoblasts fuse with the Ttn(Z)-mCherry cells of the injured host muscle and titin proteins from both cells contribute to the directionality of myofibers that is maintained along the muscle.

The online version of this article includes the following figure supplement(s) for figure 6:

**Figure supplement 1.** Titin mobility and integration upon in vivo regeneration and cell transplantation.

and work in unison along the muscle fiber. Here, we take a visual approach toward understanding sarcomere and cell biology of skeletal muscle using a fluorescent mCherry-titin fusion protein (Z-disk label) expressed at physiological levels to complement the titin-GFP fusion protein (M-band label). These animal models were specifically designed with an approach that enables us to follow titin originating from different cells in real time to study titin dynamics, sarcomere remodeling, and the mobility of titin across the syncytium after cell fusion. Visualizing opposing sarcomere integration sites

in double-heterozygous myocytes facilitates the analysis of sarcomere assembly and disassembly. We find increased mobility of Z-disk titin vs. M-band titin in FRAP experiments. These data nicely complement our earlier work on cardiomyocytes (*Rudolph et al., 2019*). Most myotubes expressed at least two titin isoforms (biphasic fit of the fluorescent recovery curve), so that skeletal muscle cells appear more homogeneous than cardiomyocytes with respect to titin isoform expression. Independent of the isoform makeup, protein exchange rates were largely similar between cardiac and skeletal muscle cells (*Rudolph et al., 2019*). Interestingly, the exchange is faster at the Z-disk than at the M-band, likely due to the integration of the newly synthesized protein with Z-disk titin mRNA available 1 hr earlier than M-band titin based on the speed of transcription (*Jonkers and Lis, 2015*). Alternatively, the contribution of short titins such as the Novex-3 isoforms, which contain the Z-disk, but not the M-band sequences, could help explain the difference.

To better understand titin distribution after cell fusion, we developed a theoretical model of titin diffusion. We assumed that titin synthesized in one region of the syncytium diffuses into adjacent areas while decaying according to its half-life. Based on published data, we estimated the half-life of titin in our system to be ~3.5 days ($\tau = 3.024 \times 10^5$ s). However, our measured diffusion coefficient for titin ($D = 0.3$ μm$^2$ s$^{-1}$) does not align with the observed width of the titin gradient (~50 μm) after 8 hr. If we maintain the experimentally determined diffusion coefficient, the half-life would need to be under 100 min, an unrealistic value. Alternatively, using the estimated half-life would require a much smaller diffusion coefficient. These inconsistencies suggest that passive diffusion alone cannot account for the observed titin spread and that active transport mechanisms, such as microtubule-dependent transport, may play a significant role.

We used the increased fluorescence of homozygous mCherry and eGFP knock-ins to study cell fusion and the outcomes of cell therapy as they allow the analysis of protein flux, compartmentalization, and the generation of functional units. Within hours after myotube fusion in cell culture, we found titin gradually distributed throughout the resulting syncytium. The spread of titin appeared to be facilitated in myotubes where mature sarcomeres had not yet formed. Nevertheless, even in myotubes that had already established a mature sarcomere structure, titin proteins of a newly fused cell were able to travel through almost half of the syncytium. Both protein and mRNA mobility contribute to the efficient distribution of titin after fusion.

Diffusion of proteins through the cytoplasm in myocytes vs. nonmuscle cells should be much more limited based on the high protein concentration in the cytoplasm and attachment to the dense cytoskeletal network within. The speed of diffusion is inversely correlated with the hydrodynamic radius of the protein (*Arrio-Dupont et al., 1996*) and packing titin in the myofilament structure limits protein diffusion even more. Microinjection of labelled dextran molecules into myotubes revealed the decrease of the diffusion coefficient with the molecular weight from 30 μm$^2$/s for a 9.5 kDa molecule to 2 μm$^2$/s for a 150 kDa molecule (*Arrio-Dupont et al., 1996*). In a similar experiment, globular proteins of different sizes were injected into isolated muscle fibers and diffusion coefficients differed depending on the fiber type likely due to differences in myofilament packing and not contraction (*Papadopoulos et al., 2000*). The distribution of titin along the myotube with about 1000 μm$^2$/hr (~0.3 μm$^2$/s; *Figure 4e*) immediately after fusion is relatively fast compared to the much smaller dextran molecules (*Papadopoulos et al., 2000*), suggesting a contribution of additional factors such as active transport (involving microtubules and the motor proteins kinesin or dynein) vs. passive diffusion.

The directed transport of mRNA to achieve proper subcellular localization is common in all types of cells and involves the interaction between Zip-code elements on the mRNA, multiple RNA-binding proteins, and motor proteins. Thus, mRNA can be distributed 60 times faster than via passive diffusion and specific localization can be achieved. Transporting mRNA is more energy efficient than transporting protein since many proteins can be translated from a single spatially organized mRNA (*Buxbaum et al., 2015*). Indeed, myocytes use the scarce sarcomeric space to accommodate ribosomes even in adult muscle (*Rudolph et al., 2019*), suggesting that sarcomeric proteins are not transported actively in mature striated muscle cells, but rather produced on site from locally translated mRNA and limited distribution by diffusion.

In our cell culture model of myotube fusion, titin protein and mRNA from adjacent cells distribute throughout the sarcoplasm. Here, titin travels faster in cells without a mature sarcomere structure. In differentiated cells, sarcomeres are built from titins originating from both parental cells, resulting in an alternating striated pattern. Still, it has remained unclear if this also applies in vivo, where fusion

events ultimately lead to large muscle fibers, which do not form in vitro (*Almada and Wagers, 2016*). To analyze how titin is distributed and integrated during regeneration and how healthy protein can be provided to diseased muscle in vivo, we used an injury model with injection of CTX into skeletal muscle (*Garry et al., 2016*) of the Ttn(Z)-mCherry mouse and transplanted Ttn(M)-eGFP myoblasts. As injury triggers the activation of the endogenous Ttn(Z)-mCherry satellite cells and their differentiation towards myocytes, myotubes, and finally fibers, the transplanted Ttn(M)-eGFP cells differentiate as well and fuse with mCherry cells and fibers. Here, we found that fluorescent titin provides a strong label to not only quantitatively follow the repopulation of injured muscle with transplanted cells, but also evaluate the generation of a functional syncytium with continued directionality of myofibers. After 3 weeks of regeneration, eGFP-positive fibers and their alternating fluorescent pattern confirmed the proper integration of donor titin. However, unlike in our tissue culture experiments, titin did not distribute throughout the fiber, but remained compartmentalized around the respective nucleus of origin. This might in part reflect the size difference between myotubes built in vitro from 2 to 10 cells and myofibers in vivo with up to hundreds of nuclei. In vitro-generated fibers retained short mRNAs close to their nucleus, whereas long mRNAs like *titin* spread through the cell (*Pinheiro et al., 2021*), consistent with the *titin* mRNA localization in our smFISH experiments in myotubes. In vivo, single-nucleus RNA sequencing (sn-RNAseq) revealed also distinct nuclear subtypes and compartments (*Kim et al., 2020*), but did not allow statements of mRNA mobility. Our data would suggest that *titin* mRNA and the derived protein can cover distances of less than one millimeter, but will not travel from its nucleus of origin throughout the myofiber of several millimeters.

Ultimately, the difference between the fusion of cultured cells with homogeneous distribution of titin vs. compartmentalization of titin from donor cell and acceptor fiber in knock-in mice confirms the importance of in vivo studies toward understanding myocyte biology and extracting clinical relevance. Our mouse cell transplantation data suggest that in myopathies compartmentalization of the therapeutic protein after fusion of a healthy cell with a diseased fiber might restrict the therapeutic effect (most prominent for the giant protein titin). To repopulate skeletal muscle with a relevant number of cells that deliver a therapeutic protein, it would therefore be beneficial to develop treatment protocols that target the early postnatal patient or consider in utero cell therapy approaches for a higher ratio of therapeutic to diseased cells and facilitated remodeling.

In conclusion, our study advances the understanding of titin dynamics in muscle biology by providing a real-time view of titin mobility following myotube fusion. These findings highlight the need for in vivo models to better understand how titin behaves during muscle regeneration and extract clinically relevant insights for therapeutic interventions in muscle diseases. The observation of compartmentalized titin distribution underscores the challenges in cell-based therapies and emphasizes the importance of early therapeutic interventions for improved outcomes.

## Methods

### Generation of titin(Z)-mCherry knock-in mice

The mCherry cDNA was inserted into titin's exon 28 (Z-disk) via a targeting vector (*Figure 1*) using standard procedures (*Radke et al., 2007*). The animals were backcrossed on a 129/S6 background after successful integration.

### Genotyping

Genomic DNA was prepared from mouse ear biopsies with the HotSHOT method (*Truett et al., 2000*). The genotypes of the titin(Z)-mCherry (primer: fwd CAGCATCATGGTAAAGGCCATCAA, rev CATTCAAATGTTGCCATGGTGTCC) and titin(M)-eGFP mice (primer: AGAACAACAAGGAAGATTCC ACA, AGATGAACTTCAGGGTCAGCTTG, TCTCAACCCACTGAGGCATA) were determined by PCR and visualized on agarose gels.

### Animal procedures

Mice were kept at the animal facility of the MDC in individually ventilated cages and a 12 hr day and night cycle with free access to food and water. All experiments involving animals were performed according to institutional guidelines and had been approved by the local authorities

(LAGeSo Berlin, Reg 0023/20). All surgeries were performed under isoflurane anesthesia, and every effort was made to minimize suffering. Strains are available upon request following institutional guidelines.

## Isolation and cultivation of primary myoblasts

For isolation of satellite cells from the titin(Z)-mCherry and titin(M)-eGFP lines, young mice (male and female) with an age of 3–4 weeks were used. Muscles from the hind limbs were collected and cut into small pieces. First digestion takes place by incubation in collagenase II (Sigma-Aldrich) for 30 min at 4°C followed by 20 min at 37°C. The second digestion step with collagenase/dispase (Roche) is performed again first for 30 min at 4°C and then for 30 min at 37°C. The digestion is stopped and the tissue homogenate is filtered with 100 µm, 70 µm, and 40 µm cell strainer. After centrifugation (1200 rpm, 10 min), the cells are resuspended in medium (Dulbecco's Modified Eagle Medium - DMEM/F12, 15% Fetal Bovine Serum - FBS, 50 µg/ml gentamicin, 1:1000 bFGF, 1:1000 Leukemia Inhibitory Factor - LIF) and were pre-plated for 1–2 hr to remove fibroblasts before they are seeded on matrigel (VWR)-coated dishes. The cells were then cultivated complete medium (+1:50 B27) under 37°C and 5% $CO_2$ and can be split or frozen.

The differentiation of the myoblast toward myotubes can be initiated by withdrawal of growth factors via changing to differentiation medium (DMEM, 5% horse serum, 1% penicillin/streptavidin).

## Cardiotoxin injury and cell transplantation

For the analysis of titin integration and distribution during in vivo regeneration, muscles of Ttn(Z)-mCherry mice were injured and myoblasts of Ttn(M)-eGFP mice were transplanted (n = 6 mice). Samples from three animals with insufficient myoblast integration were excluded from in-depth analysis. The Ttn(Z)-mCherry mice were anesthetized by isoflurane inhalation and the left TA muscle was injured by the injection of 40 µl of 10 µM CTX. Myoblasts of Ttn(M)-eGFP mice were isolated as described above and passaged two times before transplantation. 100,000 cells in 20 µl sterile PBS were injected into the left TA muscle 1 day after the injury. One mouse injected with CTX did not receive cell transplantation and served as a reference (CTX-only control). Four mice received cell transplantation without the prior injury (cells only control). Adult male mice were block randomized, based on the litter, to the experimental or control groups. Then, 21 days after the injury, mice were euthanized and the treated and untreated contralateral control TA muscles were dissected and fixed for histological analysis.

## Single-molecule in situ hypbridization (smFISH)

Cells were fixed with 2% paraformaldehyde (PFA, sterile filtered) for 10 min at room temperature followed by permeabilization with 70% ethanol overnight at 4°C. The cells were then equilibrated in washing buffer (10% formamide and 2× saline sodium citrate [SSC] buffer) for 15 min at 37°C and the hybridization of the probes (100 nM in 10% formamide and 8% dextran sulfate) with the target RNA was performed for 16 hr at 37°C. DesignReady Stellaris probe sets against mCherry (labeled with Quasar-670, # VSMF-1031-5) and GFP (labeled with Quasar-570, # VSMF-1014-5) from Biosearch Technologies were used.

After washing the cells for 30 min at 37°C, they were stained with DAPI (1:2000 in washing buffer) for 10 min at 37°C and washed with 2× SSC buffer. The imaging was performed directly on the next day to prevent degradation of the RNA.

The samples were imaged with a widefield microscope (Nikon Eclipse Ti) with narrow bandpass filter and a ×63 objective. They were excited with the Prior Lumen 200 system and the following filters were used: DAPI (Ex: 387/11, Em: 447/60, beam splitter: HC BS 409), GFP (Ex: 470/40, Em: 525/50, BS: T 495 LPXR), Quasar-570 (Ex: 534/20, Em: 572/28, BS: HC BS 552), CalFluor-610 (Ex: 580/25, Em: 625/30, BS: T 600 LPXR), and Quasar-670 (Ex: 640/30, Em: 690/50, BS: T 660 LPXR). Then, 21 z-stack images with 0.3 µm steps were taken with a pixel size of 0.22 × 0.22 µm the Andor DU888 camera. Images were processed with the Fiji (Fiji is just ImageJ) software. Background was reduced for the mRNA channels by subtraction with a Median filtered (50 px) copy of the image and z-stacks were projected with maximal intensity.

## Immunofluorescence staining

TA and EDL muscles were dissected, fixed with 4% PFA, dehydrated in 30% sucrose, and frozen in Tissue-Tek O.C.T. Cryosections of these tissues were performed with a thickness of 10 µm. The sections were permeabilized and blocked with blocking solution (10% goat serum, 0.3% Triton X 100, and 0.2% BSA in PBS) for 2 hr. Cells were fixed with 4% PFA at room temperature for 10 min and washed with PBS, followed by permeabilization and blocking as above. The incubation with the primary antibody (diluted in PBS) was performed at 4°C overnight (α-actinin [A7811, Sigma, RRID:AB_476766] 1:100, Laminin [L9393, Sigma, RRID:AB_477163] 1:100, M-cadherin [sc-81471, SantaCruz, RRID:AB_2077111] 1:50). After washing five times with PBS, cells were incubated with a fluorescent secondary antibody (diluted 1:1000 in PBS) for 2 hr at room temperature. Stained sections and cells were mounted with ProLong Gold mounting medium.

Confocal images were acquired with a laser-scanning microscope (LSM700 and LSM710, Carl Zeiss) with a Plan-Apochromat ×63/1.4oil Ph3 objective or a Plan-Apochromat ×20/0.8 M27 objective for overview images. Overview images were obtained with a pixel size of 0.447 × 0.447 µm² and high-resolution images with a voxel size of 0.081 × 0.081 × 1 µm³. Qualitative images were replicated at least three times and representative images were shown. If a quantification was done, the number of replicates is indicated in the figure legends. Line profiles were created out of the raw, unmodified images using the Fiji software and fluorescence intensity was normalized.

## Live imaging

Live imaging experiments were carried out on the DeltaVision Elite microscope (GE Healthcare) or the CSU-W1 SpinningDisk (Nikon) microscope. For the DeltaVision microscope, the ×60 oil objective (NA 1.42) was used with the FITC filter set for imaging eGFP and the A594 filter set for mCherry imaging. The ×40 objective (NA 1.15) was used for the SpinningDisk microscope and a GFP and a mCherry filter set. The pixel size was 0.1311 × 0.311 µm². The incubator of the microscopes was adjusted and equilibrated to 37°C and 5% $CO_2$ prior imaging and a humidifier was used. Cells were kept in FluoroBrite medium (plus identical supplement as during cultivation) during imaging. To avoid photo-toxicity, the laser powers were adjusted as low as possible. Usually several cells (about 10) were selected in a point list and imaged every 30–60 min for 12–16 hr at five z-stacks. To avoid shifting of the focus during the hours of imaging, the UltimateFocus option of the DeltaVision and the Perfect focus system of the SpinningDisk microscopes were used.

The imaging of fusing myotubes areas with red and green cells in close proximity was selected. Since it was expected that only a part of these cells fuse during the selected time span, many areas (about 20–30) were selected in each experiment. The progression of fusion was measured by selecting regions of interest (ROIs) based on the fluorescence intensity threshold. As a first step, the fluorescence intensity of a negative and a bright positive neighboring cell was measured and set as 0 and 100%, respectively. The fluorescence intensity value representing 20% was selected as first threshold (weak signal) and the 50% value as second threshold (strong signal). These thresholds were used to define regions with no, weak or strong signal for red and green. In the fusion process, the overlap of red and green signals of different intensities was used to assign five different types of ROIs:

> *Only red:* detectable red signal (>20%), no green signal (<20%)
> *Majority red:* strong red signal (>50%), weak green signal (20–50%)
> *Mixed:* strong red signal (>50%), strong green signal (>50%)
> *Majority green:* weak red signal (20–50%), strong green signal (>50%)
> *Only green:* no red signal (<20%), green signal (>20%)

## Fluorescence recovery after photobleaching

FRAP experiments were performed on the DeltaVision Elite microscope with the ×60 oil objective (NA 1.42). Both fluorophores of the double-heterozygous Ttn(Z)-mCherry/Ttn(M)-eGFP myotubes were photobleached with a 488 nm laser at 25% intensity for 0.1 s. A rectangular ROI covering two sarcomeres is bleached and the fluorescence recovery was followed over 14 hr with imaging every 5 min for the first 30 min, then every 30 min for another 1.5 hr and every hour for the last 12 hr. Three individual experiments with three cells each were performed. Fluorescence intensity was measured at the

respective integration sites and between it. The signal intensities were normalized to the intensities before bleaching and to the intensities of the whole cell like it is described by *Al Tanoury et al., 2010*.

$$I_{\text{frap}-\text{norm}}(t) = \frac{I_{\text{frap}}(t) - I_{\text{base}}(t)}{I_{whole}(t) - I_{base}(t)} * \frac{I_{whole-pre}}{I_{frap-pre}}$$

These normalized data were then used to fit a one-phase association curve to it with GraphPad Prism.

$$y(t) = (y_0 + M_f) * \left(1 - e^{(-K*x)}\right)$$

This curve was then used to calculate the exchange half-life, which is the time point when 50% of the maximal signal has recovered.

$$t_{1/2} = \frac{\ln(2)}{K}$$

Most of the myotubes exhibited a recovery kinetics that could be fitted better with a two-phase association curve:

$$y(t) = (y_0 + M_f) * \left(1 - e^{(-K*x)}\right)$$

This biphasic recovery is divided into a fast and a slow phase. With this formula, GraphPad Prism also calculates the percentage of the fast phase.

Independent of the type of recovery, the mobile fraction can be calculated by the fluorescence intensity at the end (when the plateau is reached) relative to the intensity at the beginning (*da Silva Lopes et al., 2011*).

$$M_f = \frac{F_{end} - F_{post}}{F_{pre} - F_{post}}$$

## Statistics

Statistical analysis was done with the GraphPad Prism software (version 5). Differences between two data sets are analyzed by *t*-test, and differences between three or more data sets with one-way ANOVA. Data affected by two factors are analyzed by two-way ANOVA and Bonferroni post-test. Bartlett's test was used to assess the equality of variance in different samples. Significances are indicated with *$p<0.05$; **$p<0.01$; ***$p<0.001$. The number of biological replicates is indicated in the respective figure legends. Normality was tested with the D'Agostino–Pearson test.

## Acknowledgements

This work was funded by the European Research Council (ERCAdv to MG) and the German Research Foundation (DFG to MG) and by DZHK (German Centre for Cardiovascular Research)-project MD3-Nanopathology (to SEL). We thank Anje Sporbert, Anca Margineanu, and the Microscope Core Facility from the MDC for support with the confocal microscopes, and Janine Fröhlich for expert technical assistance.

## Additional information

### Funding

| Funder | Grant reference number | Author |
|---|---|---|
| European Research Council | ERCAdv | Michael Gotthardt |

| Funder | Grant reference number | Author |
|---|---|---|
| Deutsche Forschungsgemeinschaft | CRC1470 + Single | Michael Gotthardt |
| Deutsches Zentrum für Herz-Kreislaufforschung | MD3-Nanopathology | Stephan E Lehnart |

The funders had no role in study design, data collection and interpretation, or the decision to submit the work for publication.

## Author contributions

Judith Hüttemeister, Data curation, Formal analysis, Investigation, Visualization, Methodology, Writing – original draft, Writing – review and editing; Franziska Rudolph, Data curation, Formal analysis, Visualization, Methodology, Writing – review and editing; Michael H Radke, Formal analysis, Investigation, Methodology, Writing – review and editing; Claudia Fink, Formal analysis, Investigation; Dhana Friedrich, Eva Wagner, Resources, Methodology; Stephan Preibisch, Resources, Supervision, Methodology; Martin Falcke, Resources, Writing – review and editing; Stephan E Lehnart, Resources, Funding acquisition, Methodology, Writing – review and editing; Michael Gotthardt, Conceptualization, Resources, Data curation, Formal analysis, Supervision, Funding acquisition, Visualization, Methodology, Writing – original draft, Project administration, Writing – review and editing

## Author ORCIDs

Judith Hüttemeister ⓘ https://orcid.org/0009-0002-8263-616X
Michael H Radke ⓘ http://orcid.org/0000-0003-0112-9917
Stephan Preibisch ⓘ https://orcid.org/0000-0002-0276-494X
Martin Falcke ⓘ https://orcid.org/0000-0001-7137-1114
Michael Gotthardt ⓘ https://orcid.org/0000-0003-1788-3172

## Ethics

All experiments involving animals were performed according to institutional guidelines and had been approved by the local authorities (LAGeSo Berlin). All surgery was performed under isoflurane anesthesia, and every effort was made to minimize suffering.

## Decision letter and Author response

Decision letter https://doi.org/10.7554/eLife.95597.sa1
Author response https://doi.org/10.7554/eLife.95597.sa2

## Additional files

### Supplementary files

• MDAR checklist

### Data availability

Figure 1, 2, 4, S1, S2 source data contain the numerical data used to generate the figures.

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
