## [Editor Report]

In this interesting study, the authors provide important insights into how titin derived from different nuclei within the syncytium is organized and integrated after cell fusion during skeletal muscle development and remodeling. This solid work elucidates the intricate process of myofilament assembly and disassembly, made possible by tracking labeled sarcomere components. The authors developed a novel mCherry titin knock-in mice with the fluorophore mCherry inserted into titin's Z-disk region to track the titin during cell fusion. The findings of the study could be important for developing therapeutic targets for diseases associated with skeletal muscle.

---

## [Decision Letter]

**Decision letter after peer review:**

Thank you for submitting your article "Visualizing sarcomere and cellular dynamics in skeletal muscle to improve cell therapies" for consideration by *eLife*. Your article has been reviewed by 3 peer reviewers, one of whom is a member of our Board of Reviewing Editors, and the evaluation has been overseen by Christopher Huang as the Senior Editor.

Essential revisions (for the authors):

1) In the manuscript, the nature of biological and technical replicates is unclear. In most in vitro and in vivo experiments, the sample size needs to be improved. The reviewers have raised concerns about the data that was not included in the final analysis. I would expect more rigor in data analysis and its interpretation, especially microscopy analysis.

2) Secondly It is vague to understand the new information the manuscript brings as the role and localization of titin in controlling stiffness of striated muscles and fine tunes contraction is known. The authors should address the concerns about the novelty.

3) The authors should quantify the microscopy images and include them for statistical analysis.

*Reviewer #1 (Recommendations for the authors):*

The manuscript needs to be revised per the comments- especially the details of experimentation need to be included. The imaging data need to be quantified. Please increase the sample size for in vitro and in vivo experiments, repeat the statistical analysis, and define the significance levels.

*Reviewer #2 (Recommendations for the authors):*

The titin protein, a large component of striated muscle, plays a crucial role in the formation of the sarcomere during muscle development. As myocytes merge, titin integrates into the sarcomere structure, creating a stable myofilament system. The authors of the present study have shed light on the intricate process of myofilament assembly and disassembly, which is made possible by tracking labeled sarcomere components. In this study, they introduced the mCherry marker into titin's Z-disk to investigate its role in skeletal muscle development and remodeling. Their findings demonstrate that the integration of titin into the sarcomere is tightly regulated, with its unexpected mobility aiding in the uniform distribution of titin post-cell fusion. This distribution is crucial for the formation and maturation of skeletal muscle syncytium. In adult mice with mCherry-labeled titin, treating muscle injuries by introducing titin-eGFP myoblasts illustrates how myocytes integrate, fuse, and contribute to a seamless myofilament system across cell boundaries. The manuscript is well written, and the study is very novel.

Comments to improve:

Can the authors determine the level of my maker/mergers in the fusion in Figures 4 and 5 to determine the quality of the cell fusion?

*Reviewer #3 (Recommendations for the authors):*

Authors indicate that knock-in of titin doesn't affect the localization. Does the knock-in of titin fused with titin express the same combination of spliced forms of titin? Can you explain in the methods section or at the beginning of the results a bit about this?

In Figure 1c, it's very hard to visualize the α-actinin staining. Can we present it as independent, self-normalized monochrome images and their overlap? It would be great if you could also provide insets of the same. Is the variability in α-actinin across the Z-disk comparable to other studies? Why does the red band look like a broadened double peak?

Do the mCh/+ and mCh/mCh mice have higher variances for the body weight ratio and heart weight-to-body weight ratio? Is that significant?

In Figure 2, what is the long lifetime and how does it differ?

Is the short phase time for FRAP different when you FRAP a volumetric GFP data? It may be better to normalize the curves with free pools of fluorescent proteins for better quantification. Figure 2 may not be the best option.

Cardiotoxin injury and cell transplantation: Samples from three animals with insufficient myoblast integration were excluded from in-depth analysis. Please articulate the exact conditions for choosing discarded versus selected animals for the analysis of titin integration and distribution during in vivo regeneration. Is such a low sample size for controls a norm in such experiments?

In the primary antibodies, α-actinin (A7811, Σ, RRID:AB_476766) was used at 1:100 dilution, Laminin (L9393, Σ, RRID:AB_477163) at 1:100 dilution, and M-Cadherin (sc-81471, SantaCruz, RRID:AB_2077111) at 1:50 dilution. After washing five times with PBS, cells were incubated with a fluorescent secondary antibody (diluted 1:1000 in PBS) for 2 hours at room temperature. Please add and specify ecolabelling was not done because actinin and cadherin seems to be monoclonal antibodies. Please reframe the sentence.

Please indicate the sampling (lateral and sampling) and laser illumination conditions for each experiment for ICC and live imaging. Line profiles can easily cause artifacts due to geometry. Can you indicate the relative spatial variability between the staining if possible?

In the Statistical analysis section, have you performed normality tests to indicate if the tests being used are consistent with the data distribution?

---

## [Author Response]

Essential revisions (for the authors):1) In the manuscript, the nature of biological and technical replicates is unclear. In most in vitro and in vivo experiments, the sample size needs to be improved. The reviewers have raised concerns about the data that was not included in the final analysis. I would expect more rigor in data analysis and its interpretation, especially microscopy analysis.

For experiments involving animals we are bound by 3R principles and German authorities require power calculations to determine sample sizes – a prerequisite for approval. We have done this for all experiments and found after completion of the experiments with the calculated n = numbers that all resulted in significant differences between groups. This includes Figure one, where our power calculation resulted in the smallest sample size of n = 3.

We have included the additional data requested. The analysis and interpretation have been revised as outlined below in the response to (2).

(2) Secondly It is vague to understand the new information the manuscript brings as the role and localization of titin in controlling stiffness of striated muscles and fine tunes contraction is known. The authors should address the concerns about the novelty.

We have expanded the introduction and discussion to better communicate that while titin’s localization within the skeletal muscle sarcomere is well-established, this represents only a static snapshot. The novelty lies in the ability to track titin dynamics over time, both in life mice and in tissue culture – and for the first time after cell fusion. Importantly, the manuscript does not focus on titin’s role in controlling stiffness and contraction, as the models we developed are specifically designed to study titin dynamics, sarcomere remodeling, and the mobility of titin across a syncytium.

3) The authors should quantify the microscopy images and include them for statistical analysis.

We have added the quantification of the smFISH experiments which had not been quantified before – as outlined in detail in the response to the reviewers.

Reviewer #1 (Recommendations for the authors):The manuscript needs to be revised per the comments- especially the details of experimentation need to be included. The imaging data need to be quantified. Please increase the sample size for in vitro and in vivo experiments, repeat the statistical analysis, and define the significance levels.

We have added remaining quantifications and explained the limitations to include additional in vivo experiments based on the experimental design and approved animal protocols.

Reviewer #2 (Recommendations for the authors):The titin protein, a large component of striated muscle, plays a crucial role in the formation of the sarcomere during muscle development. As myocytes merge, titin integrates into the sarcomere structure, creating a stable myofilament system. The authors of the present study have shed light on the intricate process of myofilament assembly and disassembly, which is made possible by tracking labeled sarcomere components. In this study, they introduced the mCherry marker into titin's Z-disk to investigate its role in skeletal muscle development and remodeling. Their findings demonstrate that the integration of titin into the sarcomere is tightly regulated, with its unexpected mobility aiding in the uniform distribution of titin post-cell fusion. This distribution is crucial for the formation and maturation of skeletal muscle syncytium. In adult mice with mCherry-labeled titin, treating muscle injuries by introducing titin-eGFP myoblasts illustrates how myocytes integrate, fuse, and contribute to a seamless myofilament system across cell boundaries. The manuscript is well written, and the study is very novel.Comments to improve:Can the authors determine the level of my maker/mergers in the fusion in Figures 4 and 5 to determine the quality of the cell fusion?

We appreciate the reviewer’s suggestion. Cell fusion was determined by the mixing of the labeled titin originated from different cells and the break-down of the connecting membrane as an obvious criterium for cell fusion. The rate of fusion events was rather low, making it difficult to measure myomaker – especially in vivo. Based on the number of nuclei after cell fusion, we suggest that cell fusion was rather heterogenous and we therefore also expect the myomaker levels to be heterogenous. The myomaker antibody from abcam is rated as poor and we do not have access to an antibody that works well for immunofluorescence in mouse tissue.

Reviewer #3 (Recommendations for the authors):Authors indicate that knock-in of titin doesn't affect the localization. Does the knock-in of titin fused with titin express the same combination of spliced forms of titin? Can you explain in the methods section or at the beginning of the results a bit about this?

In our previous work (https://doi.org/10.1073/pnas.1904385116) we integrated dsRed at the same position, where we now inserted mCherry. Since there was no effect on titin isoform expression, we expected the same for the mCherry integration. Following the reviewer’s suggestions, we performed a titin-gel analysis on WT and titin-mCherry knockin to confirm that there is no obvious change in titin isoform expression. Please find this data now included in supplement figure S1 e.

In Figure 1c, it's very hard to visualize the α-actinin staining. Can we present it as independent, self-normalized monochrome images and their overlap? It would be great if you could also provide insets of the same. Is the variability in α-actinin across the Z-disk comparable to other studies? Why does the red band look like a broadened double peak?

We improved Figure 1c (now 1d), hopefully the a-actinin staining (as monochrome) is now better visible. We also inserted a scheme of the fluorophore localization (Figure 1c) that explains the double peak of the mCherry signal at both sites of the Z-disk and the GFP at both sides of the M-band. We have a similar double band in our titin BioID animals, where the BioID is integrated at the same position as the mCherry or dsRed. The distance is increased when the sarcomere is stretched. Depending on the resolution of the microscope the bands merge to a single signal, a double peak, or two separate signals as expected.

Do the mCh/+ and mCh/mCh mice have higher variances for the body weight ratio and heart weight-to-body weight ratio? Is that significant?

Thank you for pointing this out. The Bartlett’s test for equal variances shows no significant difference between the genotypes. This information was added to Figure S1 figure legend.

In Figure 2, what is the long lifetime and how does it differ?

The long lifetime refers to the slow phase of the recovery of fluorescently labeled titin (mCherry-labeled titin at the Z-disk or GFP-labeled titin at the M-band). This slow phase represents the time required for full recovery of titin fluorescence. The titin in the slow phase recovers in 700-800 minutes (about 12 h) in the fast phase it recovers in about 2 min. 20% of the signal has a fast and 80% a slow recovery.

Is the short phase time for FRAP different when you FRAP a volumetric GFP data? It may be better to normalize the curves with free pools of fluorescent proteins for better quantification. Figure 2 may not be the best option.

The bleaching laser spread more in the z direction than in the x and y direction, so that usually the titin signal was bleached over the whole cell depth within the selected square. We could detect this, because we followed the recovery time lapse with 5 z-stacks per time point. For the data analysis we used indeed just the stack that was in focus and not the volumetric data. It is possible that the recovery is slightly faster outside the focus plane. However, this effect would be present for the GFP-labeled M-band-Titin as well as for the mCherry-labeled Z-disk Titin, because both were analyzed identically.

Data from our previous work (https://doi.org/10.1073/pnas.1904385116) in cardiomyocytes isolated from a similar mouse line suggested that the pools of not integrated titin proteins differ between the M-band and the Z-disk region, probably due to nascent proteins, shorter isoforms (e.g. Novex3) etc. Therefore, normalization to free protein could create a bias between the two titin populations.

Cardiotoxin injury and cell transplantation: Samples from three animals with insufficient myoblast integration were excluded from in-depth analysis. Please articulate the exact conditions for choosing discarded versus selected animals for the analysis of titin integration and distribution during in vivo regeneration. Is such a low sample size for controls a norm in such experiments?

The experiments for Figure 6 were a proof-of-principle, that cell fusion with transplanted myoblasts happened and titin proteins originated from graft cells is produced and integrated successfully together with host titin. Therefore, we worked with a low animal number in alliance with the local ethical guidelines regarding animal experiments (3R). In some cases, we could not detect any GFP+ cells. Since this might be due to low myoblast quality, we excluded these mice (compare response to reviewer 1). Importantly, we show 8 representative fused cells across animals and in 8 out of 8 we did not have protein exchange throughout the merged fiber.

In the primary antibodies, α-actinin (A7811, Σ, RRID:AB_476766) was used at 1:100 dilution, Laminin (L9393, Σ, RRID:AB_477163) at 1:100 dilution, and M-Cadherin (sc-81471, SantaCruz, RRID:AB_2077111) at 1:50 dilution. After washing five times with PBS, cells were incubated with a fluorescent secondary antibody (diluted 1:1000 in PBS) for 2 hours at room temperature. Please add and specify ecolabelling was not done because actinin and cadherin seems to be monoclonal antibodies. Please reframe the sentence.

The M-cadherin antibody was only used for the staining of early fusion (Figure S3 a) to detect the cell membrane. The α-actinin antibody was used for later cell fusion staining (Figure S3 b and c) to visualize the sarcomere Z-disk structure. No co-labeling with M-cadherin and α-actinin was done.

Please indicate the sampling (lateral and sampling) and laser illumination conditions for each experiment for ICC and live imaging. Line profiles can easily cause artifacts due to geometry. Can you indicate the relative spatial variability between the staining if possible?

Voxel size for live imaging is 0.1311x0.1311x1 µm³ and 0.081x0.081x1 µm³ for confocal imaging of IF stained samples. The overview images in Figure 6 have a pixel size from 0.447x0.447 µm². There is a physiological spatial variability of the sarcomeres, depending on whether the myotubes were contracted or relaxed during the time of fixation and the exact developmental state. Therefore, we expected and observed different sarcomere length within the myotubes. This is also why the line profile in Figure 2a corresponds only to the representative image to point out the faster recovery of the mCherry-labeled Z-disk Titin. The broad double-peak spacing of the Z-disk signal could be observed in many myotubes, but not always. It depends on sarcomere length and magnification.

In the Statistical analysis section, have you performed normality tests to indicate if the tests being used are consistent with the data distribution?

We performed normality tests, but we agree with the GraphPad statistical guide that the decision of when to use a parametric test and when to use a nonparametric test is a difficult one, and this decision should not be automated. We used the D’Agostino-Pearson test, which was passed in all cases (Figure 2b-d and Figure 4d-e) except for the GFP-M-band data set in Figure 2d (likely due to the single data point at >700 min). Based on the distribution of the remaining data in 2d (between 0 and 400 min) and the potential to introduce inconsistencies by applying a nonparametric test to only one data set within an experiment, we proceeded with a t-test for this data set to maintain uniformity in statistical analysis across all conditions.